# Exploring Photoreceptor Gene Expression and Seasonal Physiology in Mediterranean Swordfish (*Xiphias gladius*)

**DOI:** 10.3390/ani14223273

**Published:** 2024-11-14

**Authors:** Giorgia Gioacchini, Sara Filippi, Chiara Cardillo, Kevin De Simone, Matteo Zarantoniello, Alessia Mascoli, Oliana Carnevali, Sabrina Colella, Giulia Chemello

**Affiliations:** 1Dipartimento di Scienze della Vita e dell’Ambiente (DiSVA), Università Politecnica delle Marche, Via Brecce Bianche, 60131 Ancona, Itlay; giorgia.gioacchini@univpm.it (G.G.); filippi.sara90@gmail.com (S.F.); chiara.cardillo1996@gmail.com (C.C.); desimonekevin1@gmail.com (K.D.S.); m.zarantoniello@univpm.it (M.Z.); alessia.mascoli@iats.csic.es (A.M.); o.carnevali@univpm.it (O.C.); 2Consorzio Interuniversitario I.N.B.B., Via dei Carpegna 19, 00165 Rome, Italy; 3National Research Council (CNR), Institute for Marine Biological Resources and Biotechnology (IRBIM), Largo Fiera della Pesca 2, 60125 Ancona, Italy

**Keywords:** pelagic fish, opsin, circadian rhythm, melatonin, reproduction, metabolism, stress response, melanomacrophages, health status

## Abstract

Swordfish (*Xiphias gladius*) are top marine predators which are facing serious threats from overfishing, pollution, and climate change, especially in the Mediterranean Sea. Environmental conditions influence the levels of melatonin-related genes and opsins, which are crucial for regulating important processes like reproduction, growth, and stress responses in fish. Our findings on the expression of these genes in the livers of immature and mature female swordfish indicate a need for more research into their specific functions in different tissues.

## 1. Introduction

Swordfish (*Xiphias gladius*; Linnaeus, 1758) is a pelagic top predator species globally distributed in the Atlantic Ocean, the Indian and Pacific Oceans, and the Mediterranean Sea [1,2,3,4,5].

Overfishing, pollution, and, more recently, climate changes are threatening swordfish stocks globally, with a particular impact on reproductive performance in the Mediterranean area [6,7,8]. An increase in the distribution range limits represents one of the consequences of these adverse factors, exposing swordfish to changes in environmental parameters that regulate various physiological processes, including growth and reproduction. Additionally, as large top predators, swordfish are susceptible to accumulating high levels of contaminants over their lifespan. This condition is further exacerbated by the rising levels of contamination in highly populated areas such as the Mediterranean Sea [9]. Genetic differences [10,11] among populations from various regions (e.g., Atlantic and Mediterranean stocks) make it challenging to manage swordfish populations effectively. This complexity is further exacerbated by the need to consider how different regions could be differently affected by environmental stressors and how individuals at different life stages or in different seasons react to the same adverse condition.

Previous research evidenced with Mediterranean Sea swordfish has observed that during the reproductive season, mature females exhibit poorer health conditions compared to immature specimens, suggesting that mature females may be more susceptible to different environmental stress than immature ones due to their high energy investment in reproduction at the expense of immune response and detoxification [12]. Additionally, it has been found that during the breeding season, Mediterranean swordfish exhibit a gonadal circadian rhythm characterized by a different modulation of melatonin receptor and opsin genes (*opsin4*, *parapinopsin*, *VA opsin*, *tmt opsin*, *rho*, and *sws)* between mature and immature females [13]. In particular, a previous study [13] provided the first evidence of the expression of both visual and non-visual opsins in the swordfish ovary despite opsins having been known to be expressed in the pineal gland and eyes [14]. The authors observed an up-regulation of some of the opsins investigated in mature females with respect to immature ones during both the breeding and the non-breeding season, opening interesting questions about their potential involvement in the activation of puberty acquisition.

These differences were not evident between mature and immature individuals during the non-breeding season, highlighting the potential role of seasonality in influencing the gonadal circadian rhythm [13]. The absence of modulation during the non-reproductive period observed previously [13] raises the possible involvement of melatonin and opsins in other physiological activities through peripheral control. In addition to the ovary, another potential target organ where these genes could be expressed and involved in regulating various activities is the liver, given its role in nutrient storage and detoxification.

Melatonin, a hormone principally produced by the pineal gland and retina, plays a crucial role in regulating circadian rhythms in vertebrates. The pineal gland releases melatonin into the bloodstream in a rhythmic pattern that mirrors daily and seasonal changes in environmental factors, in particular photoperiods and temperature [15,16]. The primary role of melatonin is to synchronize key physiological and behavioral processes such as reproduction, development and growth, antioxidant regulation, and the modulation of the stress response in fish [17,18,19,20]. Melatonin’s effects are mediated by specific receptors that are part of the seven transmembrane-domain G-protein-coupled receptor family [21,22]. Melatonin receptors have been identified in peripheral tissues of fish, such as the kidney, intestine, liver, gills, muscle, and skin [23,24]. Similarly, several studies emphasized that seasonal variations in melatonin levels and melatonin receptor expression observed during different reproductive seasons may clarify the environmental impact (photoperiod and/or temperature) on fish biology. However, the exact mechanisms involved in this process and the actions of melatonin are still not well understood [25,26]. The light-mediated control of fish physiology also occurs through photoreceptors, which regulate various biological processes, including circadian entrainment, DNA repair, metabolism, and behavior [27]. Among these photoreceptors, opsins, both visual and non-visual, are other seven-pass transmembrane G-protein-coupled receptors (GPCRs) that play a key role in mediating light photoreception and inducing the transcription of light-induced genes [28].

Unlike mammals, where both visual and non-visual photoreception is primarily ocular, non-mammalian vertebrates, including fish, possess a wide range of photoreceptive areas [29]. In addition to the common expression of opsins in the brain (notably in the pineal gland) and eyes, their presence should not be excluded in other body sites. For instance, opsins have been detected in zebrafish brain, fin, gill, gut, liver, muscle, skin, and testis, and more recently in the ovary of swordfish [13,27]. There is strong evidence that the reproductive physiology of fish is regulated by opsin modifications induced by changes in photoperiod [14]. However, other physiological functions could also be mediated by opsins. Unfortunately, the only data available in the scientific literature on opsin involvement in other biological processes refer to their influence on the regulation of metabolism in mice [30].

In swordfish, the existence of a hepatic seasonal rhythm that can control the energetic and stress recovery during the non-breeding season has not been yet investigated. During this period preceding the subsequent breeding events, females may have time to restore the energy consumed in producing a large number of eggs and use this energy to properly cope with alterations in their physiological status.

The present study aimed to fill the gap in data regarding the presence and activity of certain photoreceptors and their possible involvement in normal physiological processes, such as stress response and lipid metabolism. In addition, their presence was evaluated in association with the analysis of both molecular and histological biomarkers of stress. Focus was posed on the Adriatic Sea, which has been poorly investigated or not investigated at all, despite its important role as a feeding ground for many different pelagic species, including *Xiphias gladius*.

## 2. Materials and Methods

### 2.1. Samples Collection

Swordfish female specimens were collected in September and November 2019 from the Northern-Central Adriatic Sea within the GFMC geographical subarea GSA17. Sampling was performed in collaboration with a local commercial longliner fishing fleet operating in the same area for the entire sampling period. A total of 51 females were caught (33 and 18 specimens in September and November, respectively) through longlines set in the water at the same depth at each sampling time, happening during nighttime. All 51 females were sampled after being eviscerated by the personnel on board the vessel, and for each, the total weight, gonad and liver weight were recorded. Two different portions of the liver were collected from all females and properly stored at −80 °C and in a fixative solution at +4 °C to perform molecular and histological analysis, respectively. The maturity stage of each female was determined through the histological analysis following the methods presented in a previously published study [31]. Briefly, ovaries from immature females were identified by the presence of compact ovigerous lamellae, surrounding multiple distinct oogonial nests of oocytes at the chromatin nucleolus stage. Immature females also display perinucleolar oocytes varying in number. In contrast, mature females, after the reproductive period, exhibit ovaries at the regenerating stage, characterized by oocytes of more heterogeneous sizes compared to those of immature individuals. In addition to perinucleolar oocytes, mature females also have a few larger oocytes at the early lipid stage.

### 2.2. Hepatosomatic Index (HSI)

The hepatosomatic index was calculated with the following formula:$$LW\;\left(g\right)÷TW\left(g\right)\times 100$$
where *LW* represents the liver weight expressed in grams and *TW* refers to the total weight expressed in grams.

### 2.3. Molecular Analysis

#### 2.3.1. RNA Extraction and cDNA Synthesis

Total RNA was extracted from a portion of the liver (stored at −80 °C) of both immature and mature females, respectively, and was performed using RNAzol RT reagent (Sigma-Aldrich, Milan, Italy) following the manufacturer protocol. All extracted samples were treated with the DNase I- Amplification Grade kit (Sigma-Aldrich, Milan, Italy) to remove the presence of both double-stranded and single-stranded DNA. Final RNA concentration was determined by the NanoPhotometer P-Class (Implen, München, Germany), while RNA integrity was verified by GelRedTM staining of 28S and 18S ribosomal RNA bands on 1% agarose gel. The cDNA synthesis was performed with 1 µg of total RNA using the LunaScript RT SuperMix Kit (New England Biolabs, Ipswich, MA, USA).

#### 2.3.2. Real-Time PCR

PCRs were performed using an iQ5 iCycler thermal cycler (Bio-Rad, Hercules, CA, USA). Polymerase chain reactions were set on a 96-well plate by mixing, for each sample, 1 µL cDNA diluted at a 1:10 ratio, 5 µL of 2× concentrated iQTM Sybr Green (Bio-Rad), 0.3 µM of forward primer, and 0.3 µM of the reverse primer. All the reactions followed the same thermal profile: 3 min at 95 °C and then 45 cycles of 20 s at 95 °C, 20 s at 60 °C, and 20 s at 72 °C. At the end of each cycle, fluorescence was monitored, and the melting curve analyses showed in all cases one single peak. The PCR analyses were performed to determine the relative expression of genes involved in stress response (superoxide dismutase CuZn (*sod1*), superoxide dismutase FeMn (*sod2*), and heat shock protein 4b (*hspa4b*)), and in melatonin metabolism (melatonin receptor 1b (*mel1b*), tryptophan hydroxylase (*tph1*), and acetylserotonin O-methyltransferase (*asmt*)). The relative gene expression of the visual opsins—short wavelength-sensitive opsin (*sws*) and rhodopsin (*rho*)—and the non-visual opsins (melanopsin (*opsin4*), teleost multiple-tissue opsin (*tmt*), *parapinopsin*, and vertebrate ancient opsin (*VA opsin*)) was also quantified. Actin alpha 1 skeletal muscle a (*acta1a*) and ribosomal protein L7 (*rpl7*) were selected using NormFinder (V0.963) and used as internal standards in each sample to standardize the results by eliminating variation in mRNA and cDNA quantity and quality [32]. The efficiency values of each primer were >92%, with R^2^ values within the range of 0.981 to 0.998. The sequences of the primers are listed in Table 1. No amplification products or primer–dimer formations were observed in the negative controls and control templates, respectively. The amplification products that were obtained were sequenced, and their homology was verified. The data obtained were analyzed using the iQ5 optical system software version 2.0 (Bio-Rad), which included the GeneEx Macro iQ5 Conversion and GeneEx Macro iQ5 files.

### 2.4. Histological Analysis

Liver samples, after being properly stored in a formaldehyde/glutaraldehyde solution (NaH_2_PO_4_–H_2_O + NaOH + Formaldehyde (36.5%) + Glutaraldehyde (25%) + H_2_O) (Sigma Aldrich, Milan, Itlay) at 4 °C, were processed as described by Chemello et al. (2023) [33]. Successively, samples fixed in paraffin (Bio-Optica, Milano, Italy) were cut into 5 µm sections using a microtome (Leica RM2125 RTS, Nussloch, Germany) and then stained with hematoxylin and eosin Y stain (Merck KGaA, Darmstadt, Germany). Sections were examined with an optical microscope (Zeiss Axio Imager.A2, Oberkochen, Germany) and images were acquired using a combined-color digital camera Axiocam 503 (Zeiss, Oberkochen, Germany). Liver sections were analyzed to count the number of melanomacrophages (MMs) and melanomacrophages centers (MMCs) and to calculate the lipid percentage using the ImageJ software (version 1.53t) following our previous protocol [12]. Briefly, the number of MMs was assessed in three distinct areas per liver section (3 sections per liver sample), results were presented as the mean number of MMs/area of the section (digital field area 0.1502 mm^2^), while the lipid area was analyzed in 6 sections per liver sample and described as the mean percentage of lipid area/area of the section (digital field area 0.0378 mm^2^).

### 2.5. Statistic Analysis

Statistical analyses were performed using the GraphPad Software Prism8 for Windows. GSI values, data from histological analysis (n = 22 and 29 for immature and mature females, respectively) and relative gene expression (n = 10 for both immature and mature females) were analyzed through the two-tailed unpaired t-test. A Shapiro–Wilk test was performed to assess the normal distribution of the data. Data were reported as mean values ± SD. Significance was set at *p* ≤ 0.05 for all analyses.

## 3. Results

Based on the criteria of ovary classification, 22 immature and 29 mature females were identified out of the total number of specimens sampled. All the mature females presented ovaries at the regenerating stage.

### 3.1. Biometric Parameters, HSI

Immature females were observed in the 10–15 and 15–20 kg weight classes, where they represent the total and the most abundant components, respectively (Figure 1) In particular, the 15–20 kg class consisted of 1 mature and 10 immature females. Only mature specimens showed a total weight between 20 and 40 kg (Figure 1). The analysis of the HSI showed significantly lower values in mature females than in immature ones (Figure 2).

### 3.2. Molecular Analysis

#### 3.2.1. Melatonin Receptor, Opsins, and Stress Response

Among the genes investigated through real-time PCR analysis, *tph1*, *rho*, *tmt*, and *parapinopsin* were not expressed in any of the samples analyzed, either in immature or mature females. No data graphs were produced for these genes.

The evaluation of acetylserotonin O-methyltransferase (*asmt*) and melatonin receptor (*mel1b*) mRNA relative abundance did not highlight any significant difference between immature and mature females (Figure 3A,B). Concerning the relative expression of both visual (*sws*) and non-visual opsins (*opsin4* and *VA opsin*) (Figure 3C, D, and E respectively), the only gene that significantly differed was *sws*, which was characterized by lower values in mature females (Figure 3C) (Appendix A).

#### 3.2.2. Stress and Immune Response 

The relative expression of genes involved in stress response did not highlight any significant differences between mature and immature females concerning both superoxide dismutase *sod1* and *sod2* (Figure 4A,B) (Appendix A). Differently, *hspa4b* was observed to be significantly more expressed in mature females compared to immature ones (Figure 4C).

### 3.3. Histological Analysis

#### MMCs, MMs, and Lipid Portion

Concerning the size and number of MMCs (Figure 5A,B respectively), significantly higher values were observed in mature females compared to immature specimens (Figure 6). Mature females also exhibited a significantly higher number of MMs compared to immature ones (Figure 5C). Conversely, the hepatic lipid portion was significantly lower in mature than immature females (Figure 5D), as shown by Figure 7, which shows explicative images of hepatic lipid distribution in immature (Figure 7A,B) and mature (Figure 7C,D) females (Appendix A).

## 4. Discussion

To manage the dynamic seasonal changes in the environment, animals have developed seasonal adaptive strategies for reproduction, energy storage, and migration allowing, them to adjust their physiology and behavior according to the time of year [34,35]. As a result, their annual energy allocation is optimized to ensure maximum reproductive success, while also allocating resources for survival behaviors appropriate to the season. Therefore, the ability to accurately anticipate upcoming seasons is vital for a species’ success. Animals rely on a range of environmental signals to track yearly changes and initiate the required physiological reactions. The photoperiod, or day length, is thought to regulate most seasonal behaviors, and the capacity of organisms to detect day length is called photoperiodism [36]. While the role of the photoperiod in seasonal adaptations is well-recognized, the molecular mechanisms behind this process have only recently begun to be understood. Moreover, in ectothermic species like fish, other factors apart from light could play a key role in the seasonal modulation of different physiological processes. The assessment of a peripheral control of lipid storage and health status during the non-reproductive period could enhance the understanding of swordfish stocks’ condition and provide insight into the ability of females to recover from stress conditions exacerbated by reproductive activities.

The present study investigated the expression of genes involved in peripheral circadian rhythms in the liver of swordfish females and their possible modulation between different sexual maturity stages.

In a previous study, a peripheral circadian rhythm in a swordfish ovary was demonstrated and the characterization of the transcriptional profile of opsins and key melatonin synthesis genes was proposed [13]. 

In swordfish, ovary differences were observed in the expression of the same genes between immature and mature specimens sampled during the breeding season. This modulation was not observed between mature and immature females in the non-reproductive period [13]. 

In the present study, we demonstrated the existence of a hepatic gene expression of swordfish females’ melatonin receptors (*mel1b*) and acetylserotonin O-methyltransferase (*asmt*), which is responsible for the last step of melatonin biosynthesis. Similarly, the expression of a visual opsin, (*asmt*), and two non-visual opsins (*opsin4* and *VA opsin*) were observed in females’ livers. Similarly to a previous study [13], the expression of these genes did not vary between mature and immature females, with the only exception represented by *sws*, which was less expressed in mature females compared to immature ones. These results suggested the possible involvement of the above-mentioned genes, not only in the regulation of reproduction but also in other physiological processes. However, due to a lack of knowledge regarding the molecular pathways involving *sws* in fish, the authors chose not to speculate on this result. Nonetheless, it cannot be excluded that the modulation of *sws* might affect lipid metabolism to some extent, as suggested by a study performed on mice indicating, in particular, that opsin activity could influence metabolic regulation [30]. While opsins are primarily known for their role in light perception, other factors could influence the expression of these genes, as for example, with temperatures, especially in ectothermic animals like fish. Increasing temperature leads to a rise in rhodopsin mRNA abundance in fish retina along with a redistribution of retinal fatty acid profiles [37]. This temperature-mediated modulation might indirectly influence the function and stability of rhodopsin by altering the concentration of docosahexaenoic acid (DHA), a key component of the plasma membrane in photoreceptor cells. 

In the case of swordfish, the temperature-dependent modulation of opsin genes is a concrete hypothesis, since this migratory species must adapt to environments that are very different thermal profiles, such as the Adriatic Sea (18–23 °C right after the reproductive period, reaching 8 °C during winter) and the western part of the Mediterranean (22–27 °C during the reproductive period) [38]. The detectable expression levels of the genes investigated in this study raise new questions regarding the factors regulating opsin expression specifically, as well as the need to explore the presence and role of other members of the opsin family [39]. The absence of differences in peripheral regulations between mature and immature females was observed along with the lack of significant differences in the general health conditions between immature and mature specimens. Indeed, the parameters analyzed, that could be related to a compromised condition, showed significant variations only in terms of a higher number of MMs and a higher relative expression of *hspa4b* in mature females compared to immature ones. 

Both of these findings could indicate exposure to acute-type stressors that mature females were experiencing during that period or before arriving in the Adriatic area. Their susceptibility could be related to their scarce energy reserves (highlighted by the significantly lower amount of hepatic lipid storage) consequent to reproductive activity. However, determining the cause of these patterns is challenging; for example, heat shock proteins typically respond non-specifically to various stress conditions [40]. 

By contrast, the authors attributed the higher number and size of hepatic MMCs in mature females to the aging of swordfish rather than to an acute stress response. It is well established that the size and number of melanomacrophage centers positively correlate with age and size in many fish species, as older (and consequently larger) fish have had to cope with contaminant exposure and stress conditions for a longer duration compared to younger specimens [41].

The results obtained in the present study differ from those previously observed in female swordfish during the reproductive period [12], where the evaluation of hepatic lipid fraction, both MMCs and MMs, and the different health condition of mature females appeared significantly worse than that of immature specimens.

It has already been observed that during the reproductive season, swordfish mature females may be more susceptible to different environmental stress than immature ones due to the high energy investment in reproduction at the expense of immune response and detoxification [12]. Theoretically, after the conclusion of reproductive events, females should have time to recover and return to their physiological status. In practice, their resilience capacity depends on the severity of stress and the extent to which their health status has been compromised.

In our study, although mature specimens may have experienced various stressors during the recently concluded reproductive season, they appeared to be in a phase of recovery.

## 5. Conclusions

This study investigated the expression of opsins and genes involved in melatonin biosynthesis and reception in the liver of both mature and immature female swordfish. These findings raise new interest in exploring the possible existence of peripheral circadian rhythmicity and its potential role in modulating physiological processes beyond reproduction. Of particular interest is the potential involvement of opsins in regulating basal metabolism and responses to various stressors. Given that effective conservation strategies depend on understanding species’ responses to environmental alterations, comprehending the role of opsins’ gene expression in the liver may provide insights into how swordfish perceive and adapt to environmental changes, such as those driven by climate change, which is crucial for their conservation and management.

## Figures and Tables

**Figure 1 animals-14-03273-f001:**
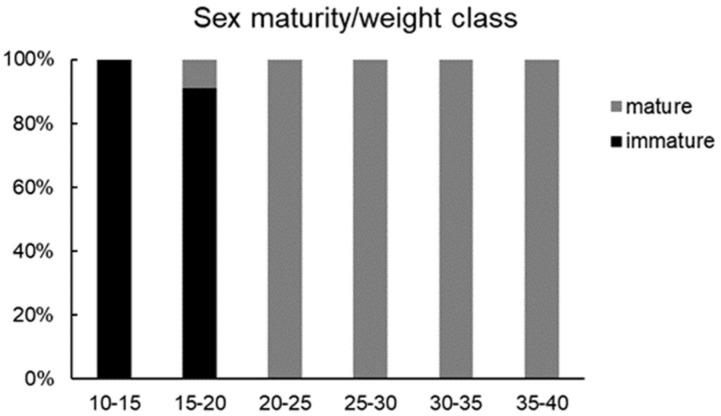
Relative frequency of mature and immature females for each size class (kg).

**Figure 2 animals-14-03273-f002:**
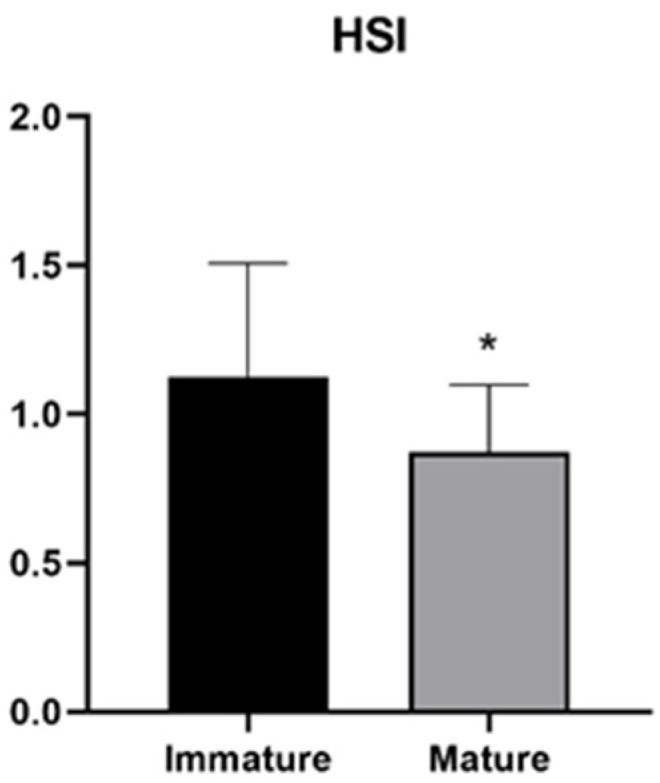
Hepatosomatic index of immature and mature swordfish females. Data reported as mean values ± SD. * *p* ≤ 0.05.

**Figure 3 animals-14-03273-f003:**
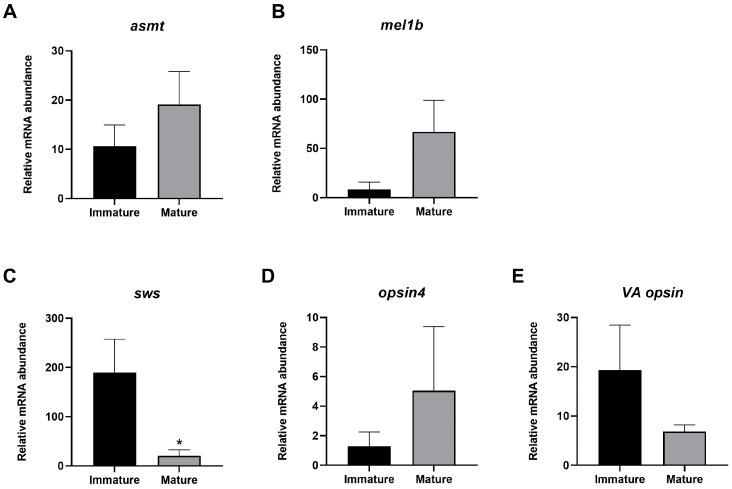
Relative gene expression of *asmt* (**A**), *mel1b* (**B**), *opsin4* (**C**), *sws* (**D**), and *VA opsin* (**E**) of immature and mature swordfish females. Data reported as mean values ± SD. * *p* ≤ 0.05.

**Figure 4 animals-14-03273-f004:**
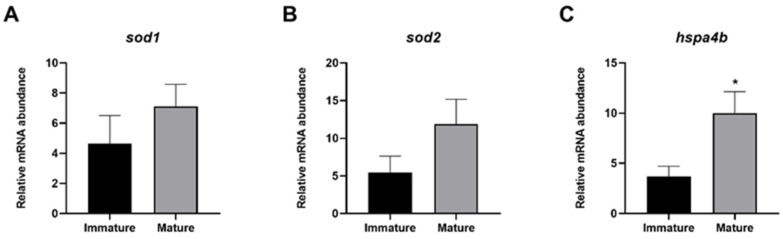
Relative gene expression of *sod1* (**A**), *sod2* (**B**), and *hspa4b* (**C**) of immature and mature swordfish females. Data reported as mean values ± SD. * *p* ≤ 0.05.

**Figure 5 animals-14-03273-f005:**
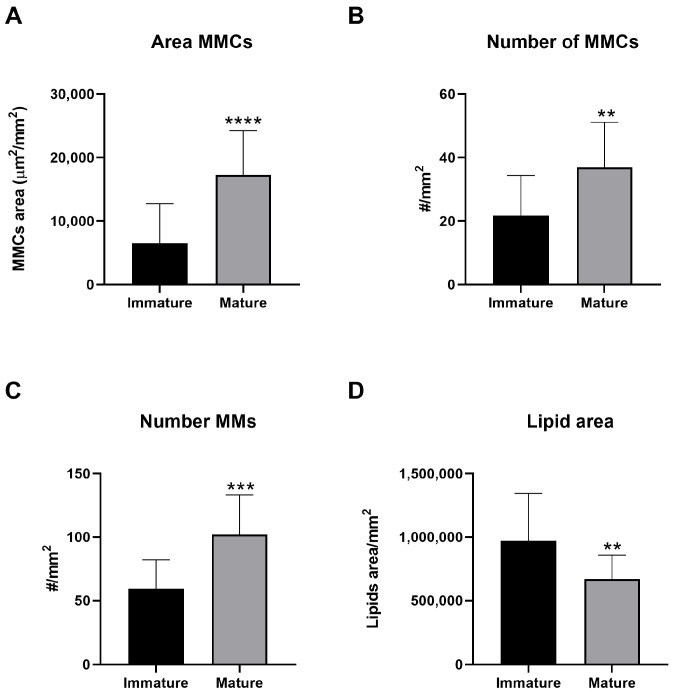
Comparison in hepatic MMC size (**A**), MMC number (**B**), MM number (**C**), and lipid portion (**D**) between immature and mature females. Data reported as mean values ± SD. ** *p* ≤ 0.01; *** *p* ≤ 0.001; **** *p* ≤ 0.0001.

**Figure 6 animals-14-03273-f006:**
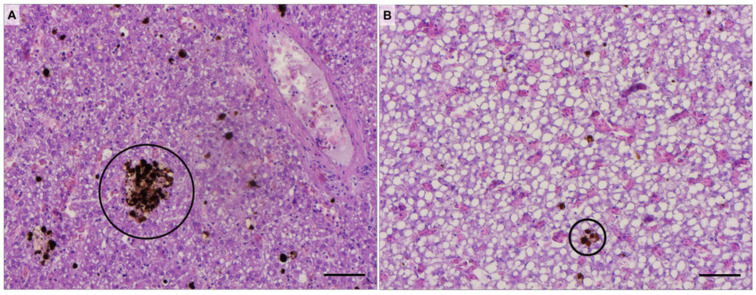
Representative histological images of liver of mature (**A**) and immature (**B**) swordfish females. Black circles identify MMCs. Scale bars: 50 µm.

**Figure 7 animals-14-03273-f007:**
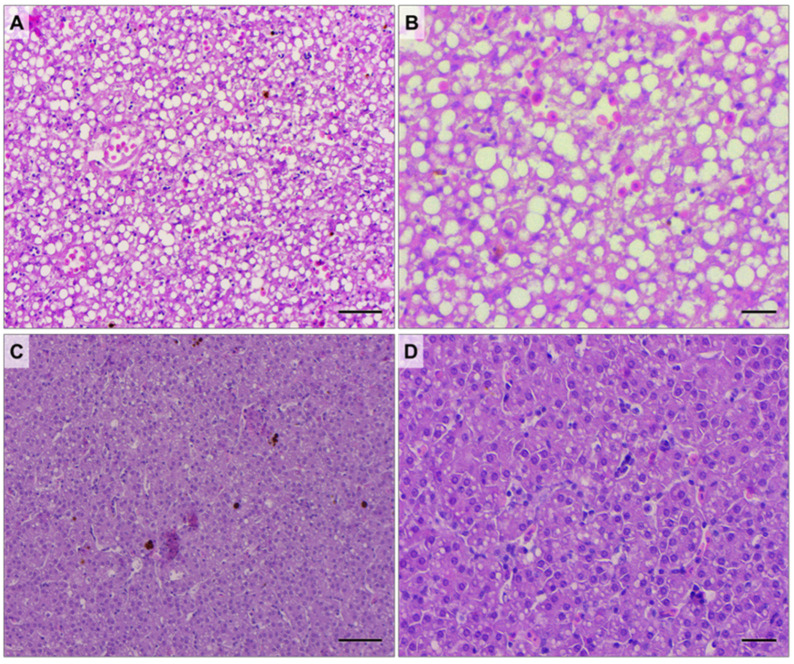
Representative histological images at different magnifications of the liver of immature (**A**,**B**) and mature (**C**,**D**) swordfish females focusing on the lipid fraction. Scale bar: 50 µm (**A**,**C**); 20 µm (**B**,**D**).

**Table 1 animals-14-03273-t001:** Gene ID and relative primer sequences.

Gene ID	Forward Primer (5′-3′)	Reverse Primer (5′-3′)
*acta1a*	TCAAAGTTCCCCTCACCGAC	GGTCTCATCGTCGTCACACA
*rpl7*	GTACTGCTCGCAAAGTGGGA	GACTTTGGGGCTGACACCAT
*sod1*	TTGGTGACCTGGGGAATGTG	GAATAGGGGCCAGTGAGGGA
*sod2*	GCGTCCATCCTGTCTTGTGAG	CGTATGTCAGGTCAGGCAGC
*hspa4b*	CAAACTGACCAACGACGCAA	ACAACGGGGTTACAAGCAGA
*asmt*	AAGACGAGTTGCCCAAAGCA	CCTTTAAGACTTTTACCTGGTGTGC
*tph1*	AGCCCCCAGATAACTCCTGT	TACATCAGGACGCGGTTAGC
*mel1b*	GGTCAGCTACTTCATGGCCT	GTCTCCGTCACAAAAAGCCG
*sws*	CTGGTCATCTGCAAGCCACT	CCAACAGTGCAAACACCCAG
*rho*	TGAATTTGGGGGAGGCTGC	TGTGTGAAGATCCACCAGGC
*opsin4*	TCAAGAGCCAGATCAAGAGCC	GTGTCCATCTGCGAAATCCCC
*tmt*	CTCAACAAACCCTGTCCGGT	CAGCGGGGTCTGTTCTGTC
*parapinopsin*	ATGCCCTCCGACAAAAGCA	TCCCCAACCAAACAGAGGTG
*VA opsin*	CTGCTGTTCGTCTGGACCTT	ACAGGTGGTTCCAATCTTGCT

## Data Availability

The raw data supporting the conclusions of this article will be made available by the authors upon request.

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
