# Peer review of "Exploring Photoreceptor Gene Expression and Seasonal Physiology in Mediterranean Swordfish (*Xiphias gladius*)"

_animals, 2024, doi:10.3390/ani14223273_

Round 1

Reviewer 1 Report

Comments and Suggestions for Authors

The authors employed quantitative RT-PCR to measure detailed expression of some circadian genes, including opsins (opsin4, VA opsin) and genes involved in melatonin biosynthesis (such as asmt) and receptor (mel1b) in the liver of both mature and immature swordfish females. Their results supported peripheral circadian rhythms and potential roles in modulating physiological processes beyond reproduction. Previous studies reported that Mediterranean swordfish exhibits a gonadal circadian rhythm characterized by a different modulation of melatonin receptor and opsin genes between mature and immature females. This manuscript is therefore a comparable report about liver circadian rhythm, which is new and interesting.

The main method is qRT-PCR, which can be improved with the advanced transcriptome sequencing in their incoming research project. In general, the overall writing of this manuscript is good; the references are appropriate, and the Figs and Tables are acceptable. On the other hand, numerous data were generated to support the main conclusions, which are consistent with the evidence and the authors addressed the main questions posed in the introduction. However, it seems that all liver samples were collected during days (although no details were provided) for these genes related to circadian rhythms. This major issue should be stated and discussed, since at least some night samples should be also collected for reasonable comparisons.

Minor issues:

Line 67: Abbreviated gene names should be italic.

In the section 2.4 Histological analysis: Provide more descriptions about how to tell the difference between mature and immature females.

Line 192: Change “HIS” to “HSI”.

Lines 187, 202, 207, 217 & 226: p should be italic.

Fig 3C: sws should be italic.

Line 319: Change “its” to “their”.

Double check the reference details. Correct at least the last one as follows: Mar. Pollut. Bull. 2022, 176, 113441.

Comments on the Quality of English Language

The overall writing of this manuscript is good.

Reviewer 2 Report

Comments and Suggestions for Authors

1.       The mRNA relative abundance of asmt and mel1b genes did not differ significantly between immature and mature females. Please provide further explanation. Is it related to the functional limitations of mel1b?

2.       Minor errors need to be fixed, such as lines 56 with extra spaces.

3.       Line 64-67 Mediterranean swordfish showed different regulatory features of melatonin receptor and opsin genes between mature and immature females. What's the difference? What's relevant to the job?

Comments on the Quality of English Language

None

Reviewer 3 Report

Comments and Suggestions for Authors

I recommend following suggestions for this article

Abstract

-Add the results in the abstract section

-Add the conclusion in the abstract

-Add the limitation of the study

Introduction

-Add how the environmental/anthropogenic stressors impact on the aquatic life, also compare the anthropogenic and natural stressors in brief by following references;

https://doi.org/10.1002/aocs.12707 , http://www.hawaii.edu/cowielab/issues.html ,

https://doi.org/10.1016/j.etap.2021.103722 , https://doi.org/10.1016/j.fsi.2020.11.021 ,

Pak. J. Bot, 53(2), 723-729

Materials and Methods

-Add a new heading for chemicals/reagents and instrumentation, used in the study

 -Add the replicate of the samples with ± SD in the statistical analysis

Results

-The obtained data/results are not added, I recommend to add the results (data).

Discussion

-Line 259, 260: “The present study investigated, for the first time, the existence of a peripheral circadian rhythm…”

I recommend not to add such statement in the study, also remove from the other part of the text

Conclusion

-Rewrite this section, it is not strongly written

References

-Replace the old references with latest one

Reviewer 4 Report

Comments and Suggestions for Authors

Referees comments on manuscript entitled ‘Exploring photoreceptor gene expression and seasonal physiology in Mediterranean swordfish (Xiphias gladius).

My expertise is in the area of fish opsins, and therefore my comments are restricted to this component of the manuscript.  

As mentioned in the earlier work from this lab (where they cited Davies et al., 2015), it is clear that fish have enormous opsin gene repertoires. The authors studied melanopsin (opn4), vertebrate ancient opsin and sws.  Why were these selected, and more importantly, what members of these subfamilies were selected?  It is likely that swordfish have four or five Opn4 genes and at least two VA opsins.  While one could argue that data from one representative of these subfamilies is interesting, it is very important to know what representatives we are looking at.  In addition, data from sws are reported.  Is this sws1 or one of several sws2 genes?  If sws1, is it known that swordfish have just one?

The connection between opsin expression in the liver and circadian rhythm left to the reader to ponder.  Are the authors suggesting that swordfish livers are light sensitive and that sws, opn4 and/or VA opsin are necessary for this light sensitivity?  What is the evidence for this?

Regarding qPCR: No information on primer efficiency has been provided. In addition, all qPCR studies are entirely dependent upon the normalizer gene expression data (the ‘internal standards’).  No data from these genes was reported.  It seems possible that expression of these internal standards might differ in mature and immature fish. Even if not, it would be good to see the data that the qPCR analyses are based on.

It is intriguing that the only opsin that was expressed at a significantly different levels in immature and mature swordfishes was sws.  If the liver was light-sensitive, would one expect that light-sensor to be tuned to short or long wavelength light?  Can sws1 function outside of the retina?  This is a ‘monostable’ opsin and in the retina, it is dependent upon enzymes in the RPE for chromophore resetting.  Can that happen in the liver?

Overall, it is not clear to me how, or why, swordfish conservation, climate change, and swordfish reproduction might be connected to opsin gene expression in the liver (i.e., a very small subset of the visual and non-visual opsin repertoire).    

“The present study investigated the existence of a peripheral circadian rhythm in the liver of swordfish females…”  I do not agree with this summary of the study.  No data on circadian rhythm were reported.  Indeed, opsin gene expression has been shown to vary during the day so it is conceivable that the sws opsin expression observations (low expression in mature fish) was a result of this sample being collected at a different time of the day compared to immature fish.

Round 2

Reviewer 1 Report

Comments and Suggestions for Authors

The authors made careful revisions in accordance with the reviewer's comments. The present version of this manuscript is acceptable for publication.

Author Response

The authors thank the reviewer for suggesting the suitability of this paper for publication in Animals following the revisions made.

Reviewer 2 Report

Comments and Suggestions for Authors

Can be accepted

Author Response

(The authors gave the same response as above.)
